# VIDHAL: Benchmarking Temporal Hallucinations in Vision LLMs

## Abstract

Vision Large Language Models (VLLMs) are widely acknowledged to be prone to hallucinations. Existing research addressing this problem has primarily been confined to image inputs, with sparse exploration of their video-based counterparts. Furthermore, current evaluation methods fail to capture nuanced errors in generated responses, which are often exacerbated by the rich spatiotemporal dynamics of videos. To address these two limitations, we introduce VIDHAL, a benchmark specially designed to evaluate video-based hallucinations in VLLMs. VIDHAL is constructed by bootstrapping video instances across a wide range of common temporal aspects. A defining feature of our benchmark lies in the careful creation of captions which represent varying levels of hallucination associated with each video. To enable fine-grained evaluation, we propose a novel caption ordering task requiring VLLMs to rank captions by hallucinatory extent. We conduct extensive experiments on VIDHAL and comprehensively evaluated a broad selection of models, including both open-source and proprietary ones such as GPT-4o. Our results uncover significant limitations in existing VLLMs with respect to video-based hallucination generation. Through our benchmark, we aim to inspire further research on i) holistic understanding of VLLM capabilities, particularly regarding hallucination, and ii) advancing VLLMs to alleviate this problem.

## 1 Introduction

Building on the advancements of Large Language Models (LLMs), Vision LLMs (VLLMs) have recently gained significant attention. Models such as LLaVA [36, 34] have shown impressive performance across various visual understanding tasks involving both images and videos. Despite their potential, VLLMs are notably prone to hallucinations, where generated responses that appear to be plausible contradict the visual context [1, 59]. This problem significantly compromises the reliability of VLLMs, hindering their practical use in real-world applications.

To tackle this challenge, some methods propose to leverage post-hoc techniques such as contrastive decoding [22, 77, 11, 78] and attention calibration [16, 41, 39, 66, 14, 71, 58]. Other efforts have been devoted to the evaluation of hallucinations in VLLMs. For example, CHAIR [47] initially studies object-based hallucination evaluation with the aid of the image captioning task. Subsequent studies [31, 38, 20, 10] instead harness paired ⟨*positive, hallucinatory*⟩ questions to probe such hallucinations. Additionally, MMHalBench [50] and AMBER [53] expand beyond object-based evaluations by constructing benchmarks that cover attribute and relationship hallucinations.

Unlike their image-based counterparts, video hallucinations pose unique challenges primarily due to the intricate spatiotemporal dynamics of videos [29, 45, 6, 12, 40, 42]. In particular, video-specific temporal aspects, such as movement direction and chronological order of events, are especially concerning for video-based VLLMs. Furthermore, the richness of video content necessitates a finer-

grained understanding, making VLLMs more vulnerable to nuanced hallucinations. Nonetheless, to the best of our knowledge, video-based hallucinations remain underexplored in the existing literature.

To address this research gap, we present VIDHAL, a benchmark specifically designed to evaluate video-based hallucinations of VLLMs. VIDHAL features videos that comprehensively cover a broad range of temporal aspects, such as entity actions and sequence of events. Each video is automatically annotated with multiple captions exhibiting *varying levels* of aspect-specific hallucinations, capturing both subtle and significant discrepancies. In addition, we perform detailed human validation to ensure the robustness and reliability of our annotation process. An additional motivation stems from the limited metrics for quantifying hallucinations in VLLMs. To capture fine-grained hallucinatory errors of these models, we propose a unique caption ordering task that requires models to rank captions by hallucination levels. This consequently leads to a ranking-based NDCG metric and an MCQA accuracy metric, both are distinct from prior ones and specifically tailored to evaluate nuanced hallucinations in video-based VLLMs.

Using our VIDHAL dataset, we benchmark thirteen VLLMs including both open-sourced and proprietary models, with abstracted results summarized in Figure 1. Through these extensive experiments, we identify limitations in nuanced video understanding among all evaluated VLLMs. Specifically, our findings reveal that existing VLLMs struggle to differentiate between captions with varying levels of hallucination. This deficiency is particularly evident when evaluating video-specific aspects, such as *Direction* and *Order*, as illustrated in Figure 1, indicating substantial room for improvement in current video-based VLLMs. Additionally, proprietary models, *e.g.*, GPT-4o [43], often outperform open-source counterparts by significant margins.

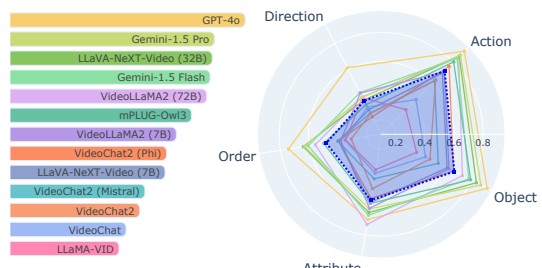

Figure 1: Multiple-Choice Question Answering (MCQA) performance of representative VLLMs on our VIDHAL benchmark. (Left) Overall ranking of VLLMs. (Right) Detailed accuracy results pertaining to each temporal aspect, wherein higher scores indicate fewer hallucinations.

Overall, the contributions of this work are three-fold:

• We present VIDHAL, a benchmark dataset dedicated to video-based hallucination evaluation of VLLMs. Our dataset is distinguished by i) video instances sourced from public video understanding datasets encompassing a diverse range of temporal concepts and ii) captions with varying levels of hallucination[1].

• We introduce a novel evaluation task of caption ordering along with two metrics designed to evaluate fine-grained hallucination generation in existing VLLMs.

• We conduct extensive experiments on VIDHAL with a variety of VLLMs, uncovering limitations in their fine-grained video reasoning abilities, particularly in their tendency to generate hallucinations.

## 2 Related Work

**Vision Large Language Models.** The emergence of powerful LLMs has advanced the development of VLLMs [36, 34, 25, 9, 62, 63, 61]. Typical methods in this category include LLaVA [36], mPLUG-Owl [63, 61, 62], InstructBLIP [9], and MiniGPT-4 [75]. These VLLMs rely on aligning vision encoders with LLMs using connective modules such as Q-Former [9, 26, 25, 67, 8] or MLPs [36, 34, 49] with the instruction tuning stage. Recent methods have extended visual inputs from images to (long) videos, delivering impressive joint spatial-temporal reasoning capabilities. For instance, VideoLLaMA2 [8] enhances the LLaMA model with video understanding capabilities through a Spatial-Temporal Convolution (STC) module. LLaVA-NeXT-Video [35, 68] presents an AnyRes approach that enables reasoning with long videos.

**Hallucinations in VLLMs.** Despite their impressive performance on visual reasoning benchmarks, current VLLMs remain notoriously susceptible to hallucinations [18, 39, 76, 5]. A common demonstration is that the generated responses contain information which is inconsistent with the visual

---

[1]Our VIDHAL dataset will be made available to the public.

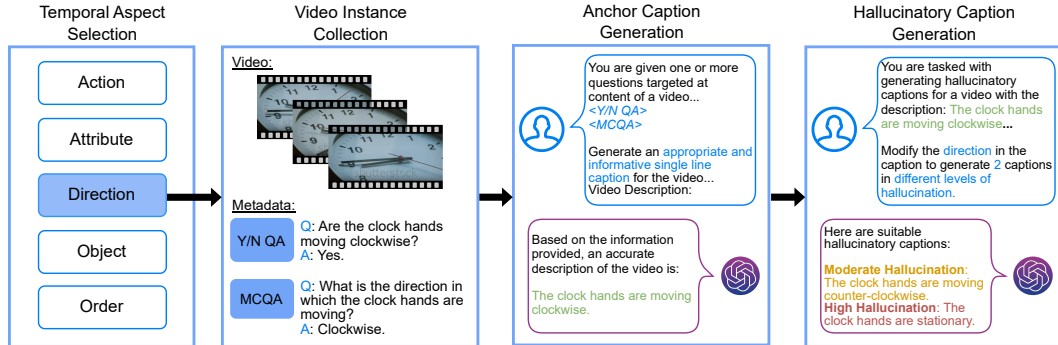

Figure 2: Overview of our VIDHAL benchmark construction pipeline. Using *direction* as an example from the five selected aspects, we begin by sourcing relevant video instances from existing datasets. Next, the anchor (positive) caption is generated from the original video metadata. Finally, GPT-4o is employed to generate hallucinatory captions at varying levels.

content [1, 33, 65, 57]. Most approaches address the hallucination problem with post-hoc techniques. For example, LURE [73] and Woodpecker [64] develop pipelines that assist VLLMs in revising their responses using expert models. To reduce bias from unimodal and statistical priors, contrastive decoding methods, such as VCD [22] and M3ID [11], along with attention calibration techniques like OPERA [16] are employed to refine token predictions. Building on the success of reinforcement learning for preference optimization in LLM development [44], HA-DPO [69], POVID [72] and CSR [74] adopt this paradigm to fine-tune VLLMs, yielding outputs with fewer hallucinations.

**Video Reasoning Benchmarks.** The rise of video-based VLLMs has driven the development of numerous video benchmarks. Notable examples, such as SEEDBench [23], VideoBench [42], MVBench [29], and VideoMME [12], focus on dynamic events requiring temporal reasoning beyond individual frames. However, these benchmarks often lack diversity in reasoning tasks and visual concepts. To address this, AutoEval-Video [6] and Perception Test [45] introduce complex reasoning tasks such as counterfactual and explanatory reasoning, while TempCompass [40] expands temporal concept coverage. Several benchmarks [31, 53, 50, 20, 32, 19, 55, 70, 5, 4, 56, 51, 7] have been constructed to quantify visual hallucinations, primarily targeting object-based hallucinations in images. HallusionBench [15], VideoCon [2], and Vript [60] provides partial coverage of video-based hallucinations, while VidHalluc [24] and VideoHallucer [54] introduces a benchmark for hallucination detection in videos. However, these benchmarks provide limited coverage of spatio-temporal concepts, focusing on conventional aspects like actions while neglecting other video-centric elements such as direction. Additionally, their evaluation strategies primarily follow image-based approaches, which we argue are less effective in capturing nuanced, video-specific hallucinations.

## 3 VIDHAL Dataset Construction

We introduce VIDHAL, a unique video-language benchmark designed to evaluate hallucinations of Video-LLMs in a comprehensive manner. As depicted in Figure 2, VIDHAL comprises of video instances which span a diverse spectrum of temporal aspects, including previously unexplored aspects such as directional movement. In contrast to previous studies on video hallucination evaluation [60, 54, 2], VIDHAL incorporates multiple hallucinated captions per video, enabling the assessment of video hallucinations at multiple levels of granularity.

### 3.1 Temporal Hallucinations in Videos

Hallucinations in VLLMs occur when the model fabricates details in its responses that contradict the provided visual content. Compared to images, video hallucinations extend beyond static visual elements to include misperceptions of dynamic changes within scenes. We categorize these temporal hallucinations into two semantic levels:

**Lexical Semantics (L-Sem)** captures instances where VLLMs misinterpret words related to temporal features, including nouns referring to objects or attributes (e.g., misidentifying a color change from

green to red as green to orange) and verbs describing actions (e.g., interpreting "kicking a ball" as "throwing a ball").

**Clause Semantics (C-Sem)** encompasses errors involving event descriptions and their sequences, where the VLLM incorrectly predicts the order of events occurring in the video. For example, given sequentially occurring events $A$ and $B$ in a video, the model may perceive $B$ preceding $A$.

By addressing these two dimensions of video-based hallucinations, VIDHAL offers holistic coverage over the level of detail in which VLLMs may hallucinate.

## 3.2 Temporal Concept Selection

Prior research on hallucination evaluation for both images [31, 53, 47] and videos [54, 60, 15] has predominantly focused on common visual aspects such as action- and object-based hallucinations. However, video-based hallucinations may involve additional dynamic factors associated with spatio-temporal patterns, which these studies overlook. In light of this, we propose to focus on the following five aspects to ensure comprehensive coverage of temporal concepts. Specifically, the first four aspects address hallucinations based on lexical semantics, while the fifth targets clause semantics.

- **Attribute (L-Sem)** describes the fine-grained characteristics and properties of objects or subjects in the video. We additionally categorize this aspect into sub-aspects of *Size*, *Shape*, *Color*, *Count* and *State Change*.

- **Object (L-Sem)** relates to the interactions between objects and entities within the video. We further delineate this aspect into two fine-grained sub-aspects: *Object Recognition*, identifying the objects engaged in interactions, and *Interaction Classification* which concentrate on how these objects interact with other objects or subjects.

- **Action (L-Sem)** refers to the movements and behaviours exhibited by entities.

- **Direction (L-Sem)** indicates the orientation and movement trajectory of subjects or objects.

- **Event Order (C-Sem)** represents the correct sequence of events in the video. During our collection, we retain videos that contain at least three distinct events.

We present an example that illustrates the direction aspect in Figure 2, with additional examples available in the supplementary material.

## 3.3 Hallucinatory Caption Generation

Based on the aspects in Section 3.2, we build our benchmark upon four public video understanding datasets: TempCompass [40], Perception Test [45], MVBench [29] and AutoEval-Video [6]. TempCompass and MVBench extensively cover all five temporal aspects, while Perception Test and AutoEval-Video highlights human-object interactions and attribute changes, respectively.

Existing hallucination benchmarks [31, 53] rely mostly on binary questions for evaluation, limiting their efficacy in detecting subtle video hallucinations, such as minor event inconsistencies. To address this issue, we advocate a novel evaluation protocol incorporating several carefully annotated captions. Specifically, each video will be annotated with a set of $M$ captions that reflect varying degrees of hallucination in VLLMs. Given the cost and labor intensity of manual annotation, we follow existing studies such as PhD [38] and MVBench [29], opting for automatic caption generation using a carefully designed pipeline illustrated in Figure 2.

**Anchor Caption Generation.** The video instances in VIDHAL are sourced from various public datasets, resulting in distinct associated metadata such as long-form captions in AutoEval-Video and question-answer pairs in MVBench. To ensure structure consistency and information granularity in the respective dataset description across all instances, we automatically generate an anchor caption for each video. Specifically, we input the metadata for each video $V^i$ into GPT-4o and prompt it to generate a concise and accurate description $y^i_+$ using the provided metadata information.

**Hallucinatory Caption Generation.** After obtaining the positive caption for each video instance, we augment the dataset with $M - 1$ additional captions containing hallucinated content. For a given video instance $V^i$, we construct a set $\mathcal{Y}^i_- = \{y^{i,1}_-, \cdots, y^{i,M-1}_-\}$ containing captions with different

| | Dataset | Temporal Aspects | | | | | | | | | | Task Formats | Evaluation Metrics |
|---|---|---|---|---|---|---|---|---|---|---|---|---|---|---|
| | | Action | Attribute | | | | | Direction | Object | | Order | | |
| | | | Size | Shape | Color | Count | State-Change | | Recognition | Interaction | | | |
| Video Reasoning | SEEDBench [23] | ✓ | ✗ | ✗ | ✗ | ✗ | ✗ | ✗ | ✗ | ✗ | ✓ | MCQA | Accuracy |
| | VideoBench [42] | ✓ | ✓ | ✓ | ✓ | ✓ | ✗ | ✗ | ✓ | ✗ | ✗ | MCQA | Accuracy |
| | MVBench [29] | ✓ | ✗ | ✗ | ✗ | ✗ | ✗ | ✓ | ✓ | ✗ | ✓ | MCQA | Accuracy |
| | Video-MME [12] | ✓ | ✓ | ✓ | ✓ | ✓ | ✗ | ✗ | ✓ | ✗ | ✗ | MCQA | Accuracy |
| Hallucination Evaluation | Vript [60] | ✓ | ✗ | ✗ | ✗ | ✗ | ✗ | ✗ | ✓ | ✓ | ✓ | Video Captioning / Event Ordering | F1 Score / Accuracy |
| | VideoCon [2] | ✓ | ✓ | ✓ | ✓ | ✓ | ✗ | ✗ | ✓ | ✗ | ✓ | VL Entailment | ROC-AUC |
| | HallusionBench [15] | ✓ | ✗ | ✗ | ✗ | ✗ | ✗ | ✓ | ✗ | ✗ | ✓ | Y/N QA | Accuracy |
| | VIDHAL (Ours) | ✓ | ✓ | ✓ | ✓ | ✓ | ✓ | ✓ | ✓ | ✓ | ✓ | MCQA / Caption Ordering | Accuracy / NDCG |

Table 1: Comparison of our benchmark dataset with existing video-based reasoning and hallucination evaluation datasets. For datasets with multiple evaluation tasks, only those relevant to hallucination evaluation are included. VL Entailment denotes the task of *video-language entailment*, while *Event Ordering* prompts the model to determine the chronological sequence of scenes in a video.

levels of hallucination based on the temporal concepts associated with it. Specifically, $y_-^{i,k}$ exhibits heavier hallucination than $y_-^{i,j}$ for $j < k$. We leverage GPT-4o to generate $\mathcal{Y}_-^i$ by combining the anchor caption $y_+^i$ and prompting it to create $y_-^{i,1}, \cdots, y_-^{i,M-1}$ progressively in increasing levels of hallucination. The set of captions associated with $V^i$ is then defined as $\mathcal{Y}^i \leftarrow \{y_+^i\} \bigcup \mathcal{Y}_-^i$ consisting of both the anchor and hallucinatory captions.

## 3.4  Dataset Statistics and Human Validation

Our VIDHAL benchmark consists of a total of 1,000 video instances. Using our automatic annotation pipeline, each video instance is tagged with $M = 3$ captions. As shown in Table 1, our VIDHAL dataset stands out from other video understanding [23, 42, 29, 12] and hallucination benchmarks [2, 15, 37] in terms of two dimensions: I) VIDHAL encompasses a diverse range of video-centric temporal aspects; and II) We introduce a novel caption ordering task along with two tailored metrics to capture subtle hallucinations previously ignored by paired questions.

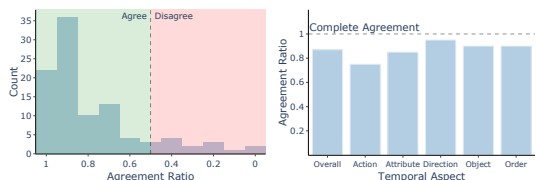

Figure 3: Human agreement on hallucination levels in the VIDHAL dataset. (Left) Distribution of agreement ratios per video sample. (Right) Average agreement ratio for each temporal aspect, with an overall average of 87%.

To ensure the reliability of our generated captions at varying levels, we randomly selected 100 examples for human validation, where each sample is labeled by 15 annotators on average. Our human validation process focuses on verifying that the order of hallucinatory captions generated by our pipeline aligns with human judgment. Figure 3 reflects an overall agreement rate of 87% between our automatically generated hallucinatory captions and human annotators, indicating consistency between these two across all temporal aspects.

## 4  VIDHAL Evaluation Protocol

Aiming to address the limitations of binary question-based benchmarks, we propose two evaluation tasks: *multiple-choice question answering* and a novel *caption ordering task*, detailed in Section 4.1. We also develop corresponding metrics to comprehensively measure hallucinations in video-based VLLMs, elaborated further in Section 4.2.

### 4.1  Evaluation Tasks

**Multiple-Choice Question Answering (MCQA)** assesses the model's spatiotemporal understanding in a coarse-grained manner. Specifically, the model is provided with a video $V^i$ and its corresponding set of captions $\mathcal{Y}^i$ as answer options. The VLLM is then instructed to select the most appropriate caption for the video.

**Caption Ordering** evaluates a model's visual reasoning from a nuanced granularity, instructing VLLMs to order the provided captions based on their hallucination level. Through pairwise comparisons across all captions, this task identifies cases where the model struggles to distinguish varying levels of hallucination severity beyond anchor-hallucination distinctions.

Specifically, we design two caption ordering sub-tasks. The first, *naive caption ordering*, requires VLLMs to rank all captions at once. However, this sub-task can confuse several VLLMs due to its inherently challenging nature and the inferior instruction-following capabilities of some models. As a complement, we propose an additional sub-task, *relative caption ordering*, which decomposes the prior task into multiple paired caption ordering tasks. Since each paired ordering task is answered in isolation, the VLLM may

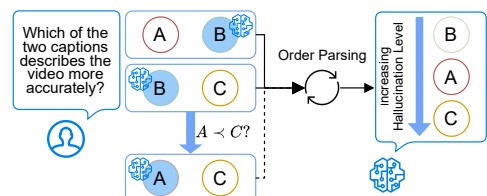

Figure 4: Visual illustration of *relative caption ordering* task in VIDHAL.

produce a non-transitive, cyclic ranking. To circumvent this, we query the model with consecutive caption pairs, prompting the final pair only if multiple orderings are possible. For instance, given captions $A$, $B$, and $C$, if the model predicts $A \prec B$ and $B \prec C$, the overall order $A \prec B \prec C$ can be directly inferred. However, if it instead ranks $B \prec A$, as shown in Figure 4, we additionally include a third comparison between $A$ and $C$ to resolve any ambiguity in determining in the final order.

Notably, our relative caption ordering task is more challenging than previous binary questions. This complexity arises from certain paired questions in VIDHAL where both options are hallucinatory, making them harder to distinguish as opposed to ⟨*positive, hallucinatory*⟩ pairs.

### 4.2 Evaluation Metrics

**Notations** For a particular video instance $V^i$, we define the ground truth caption order for $V^i$ to be $\mathcal{Y}_*^i = (y_+^i, y_-^{i,1}, \cdots, y_-^{i,M-1})$. Further let the $j^{th}$ element in this ordering be indexed as $\mathcal{Y}_*^{i,j}$.

**MCQA** We employ the standard accuracy metric:

$$\text{Accuracy} = \frac{1}{N} \sum_{i=1}^{N} \mathbb{I} \left[ R_{MCQA}(V^i, \mathcal{Y}^i) = y_+^i \right], \tag{1}$$

where $N$ is the number of video instances, $\mathbb{I}$ denotes the indicator function, and $R_{MCQA}(V^i, \mathcal{Y}^i)$ represents the best matched caption from $\mathcal{Y}^i$ for $V^i$ as predicted by a VLLM.

**Caption Ranking** Inspired by metrics from the information retrieval domain [13], we adapt the well-established Normalized Discounted Cumulative Gain (NDCG) [17] for hallucination assessment in VIDHAL. Unlike previous metrics like POPE [31], our metric awards partial credit for correctly ordered caption pairs even when the optimal ranking is not achieved. As such, we expect the metric to effectively capture and distinguish both subtle and severe hallucinations generated by video-based VLLMs. Formally, we define our adapted NDCG metric as follows:

$$\text{NDCG} = \frac{1}{N} \sum_{i=1}^{N} \frac{\text{DCG}_i - \text{rDCG}_i}{\text{iDCG}_i - \text{rDCG}_i}, \tag{2}$$

where $\text{DCG}_i$ is formulated as:

$$\text{DCG}_i = \sum_{j=1}^{M} \frac{r\left(\hat{y}^{i,j}, \mathcal{Y}_*^i\right)}{\log(j+1)}, \tag{3}$$

and $\hat{y}^{i,j}$ represents $j^{th}$ caption in the ranked order predicted by the VLLM. The perfect ordering is achieved when $\hat{y}^{i,1} = y_+^i$ and $\{\hat{y}^{i,j} = y_-^{i,j-1}\}_{j=2 \to M}$. To evaluate predicted caption orders relative to this ideal sequence, a relevance function $r\left(\hat{y}^{i,j}, \mathcal{Y}_*^i\right)$ is designed to assign higher scores to $\hat{y}^{i,j}$ with lower hallucinatory extent.

$$r(\hat{y}^{i,j}, \mathcal{Y}_*^i) = M + 1 - \text{pos}(\hat{y}^{i,j}, \mathcal{Y}_*^i), \tag{4}$$

| Model | Vision Encoder | LLM | #Params | #Frames | Accuracy | NDCG | |
|-------|----------------|-----|---------|---------|----------|------|---|
| | | | | | | Naive | Relative |
| *Baseline* | | | | | | | |
| Random | - | - | - | - | 0.326 | 0.505 | 0.480 |
| *Open-Sourced Models* | | | | | | | |
| VideoChat [28] | EVA-CLIP-G | Vicuna | 7B | 8 | 0.381 | 0.475 | 0.488 |
| LLaMA-VID [30] | EVA-CLIP-G | Vicuna | 7B | 1fps | 0.358 | 0.486 | 0.521 |
| VideoChat2 (Vicuna) [29] | UMT-L | Vicuna | 7B | 16 | 0.426 | 0.486 | 0.577 |
| VideoChat2 (Mistral) | UMT-L | Mistral | 7B | 16 | 0.443 | 0.503 | 0.475 |
| VideoChat2 (Phi) | UMT-L | Phi3 | 3.8B | 16 | 0.514 | 0.626 | 0.612 |
| mPLUG-Owl3 [61] | SigLIP/SO400M | Qwen2 | 7B | 16 | 0.596 | 0.641 | 0.707 |
| LLaVA-NeXT-Video (7B) [68] | SigLIP/SO400M | Vicuna | 7B | 32 | 0.509 | 0.518 | 0.620 |
| LLaVA-NeXT-Video (32B) | SigLIP/SO400M | Qwen1.5 | 32B | 32 | **0.663** | 0.641 | 0.747 |
| VideoLLaMA2 (7B) [8] | CLIP ViT-L/14 | Mistral | 7B | 8 | 0.541 | 0.564 | 0.622 |
| VideoLLaMA2 (72B) | CLIP ViT-L/14 | Qwen2 | 72B | 8 | 0.647 | **0.787** | **0.760** |
| *Proprietary Models* | | | | | | | |
| GPT-4o [43] | - | - | - | 1fps | 0.772 | 0.840 | 0.826 |
| Gemini-1.5 (Flash) [46] | - | - | - | 1fps | 0.657 | 0.738 | 0.745 |
| Gemini-1.5 (Pro) | - | - | - | 1fps | 0.671 | 0.765 | 0.753 |

Table 2: Benchmark performance of VLLMs on our VIDHAL dataset. #Params refers to the number of parameters of the base LLM used. The best performance for each task is highlighted in **bold** for open-sourced models, and underlined for closed-sourced models.

where $\text{pos}(\hat{y}^{i,j}, \mathcal{Y}_*^i)$ denotes the position of $\hat{y}^{i,j}$ in $\mathcal{Y}_*^i$. Finally, $\text{DCG}_i$ is normalized to a range of $[0, 1]$ using $\text{iDCG}_i$ and $\text{rDCG}_i$, with a score of 1 indicating perfect alignment of the predicted order with $\mathcal{Y}_*^i$. Specifically, these terms represent the maximum and minimum $\text{DCG}_i$ scores obtained from the optimal ordering $\mathcal{Y}_*^i$ and its reverse, respectively,

$$\text{iDCG}_i = \sum_{j=1}^{M} \frac{r\left(\mathcal{Y}_*^{i,j}, \mathcal{Y}_*^i\right)}{\log(j+1)}, \ \text{rDCG}_i = \sum_{j=1}^{M} \frac{r\left(\mathcal{Y}_*^{i,M-j}, \mathcal{Y}_*^i\right)}{\log(j+1)}. \tag{5}$$

## 5 Experiments

### 5.1 Experimental Settings

**Models.** We evaluated thirteen VLLMs from eight different model families, including six open-source models: VideoChat [28], LLaMA-VID [30], VideoLLaMA2 [8], VideoChat2 [29], mPLUG-Owl3 [61] and LLaVA-NeXT-Video [68], and two proprietary models: GPT-4o [43] and Gemini-1.5 [46]. These models represent a wide variety of architectural designs and training paradigms. Additionally, we included a random baseline that selects and ranks candidate options randomly.

**Implementation Details.** All experiments were conducted using four NVIDIA A100 40GB GPUs. The input captions in $\mathcal{Y}^i$ were presented in a randomized order using a fixed, predefined randomization seed across experiments. We adhered to the inference and model hyperparameters outlined in the respective original models, and employed greedy decoding during generation for a fair comparison.

### 5.2 Overall Results

**Benchmark Results.** We present the overall results of representative VLLMs in Table 2 across both MCQA and caption ordering tasks. We make three key observations from this table: i) Proprietary models demonstrate superior results compared to open-sourced models. In particular, GPT-4o achieves the best performance on all tasks, surpassing other models by significant margins. ii) Larger VLLMs generally outperform smaller ones in both tasks. This result is supported by the comparison of different LLM bases for the VideoLLaMA2 and LLaVA-NeXT-Video models. iii) The caption ordering task poses greater difficulty for current VLLMs than MCQA, evidenced by the larger performance margins between the VLLM models and the random baseline. Notably, VideoChat and VideoChat2 (Mistral) show slight to no improvement over the random baseline across both caption ordering tasks. This indicates that current VLLMs greatly suffer from poor fine-grained video understanding and are inclined to generate hallucinations.

**Aspect-aware Results.** Figure 5 highlights the fine-grained, aspect-specific performance of the evaluated models. Notably, VLLMs demonstrate substantially stronger results on the *Action* and *Object* aspects compared to others. This can likely be attributed to current visual instruction tuning datasets predominantly emphasizing object-centric recognition and coarse-grained activity classification, potentially encouraging strong reliance on image-based priors when generating predictions. In contrast, these models tend to underperform on temporally nuanced aspects such as direction and event order, which are inherently unique to the video modality.

We further analyzed the distribution of results for the relative caption ranking task across sub-aspects of the *Attribute* and *Object* aspects in Figure 6. While VLLMs generally maintain consistent performance across *Attribute* sub-aspects, their effectiveness declines slightly when reasoning about *Count* and *Color*, suggesting that reasoning over such fine-grained visual properties remains challenging for VLLMs. For the *Object* aspect, several models performed significantly worse in *Interaction Classification* than in *Object Recognition*, highlighting the need to better model object interactions to bridge the gap between recognition and understanding.

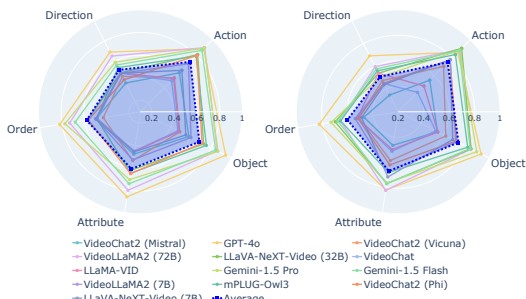

Figure 5: Aspect-specific NDCG scores for the (Left) naive and (Right) relative caption ordering.

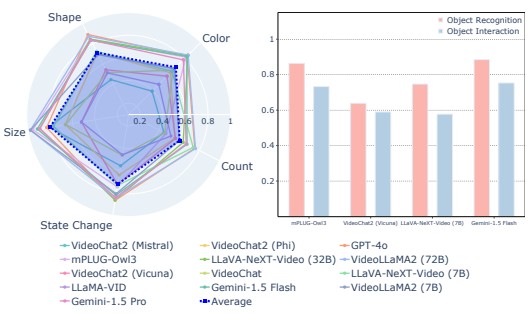

Figure 6: NDCG scores for *Attribute* (Left) and *Object* (Right) sub-aspects in caption ordering.

## 5.3 Ablation Studies

**Hallucination Differentiation Sensitivity.** We investigate the tendency of VLLMs to favor captions with higher hallucination over those with lower degree in the relative caption ranking task. For two captions with different hallucination levels $j, k$ where $j > k$, we introduce the following metric to quantify such *hallucination misalignment* cases:

$$HM_{j \to k} = \frac{1}{N} \sum_{i=1}^{N} \mathbb{I}\left[\mathcal{Y}_*^{i,j} \prec \mathcal{Y}_*^{i,k}\right]. \tag{6}$$

which reflects the proportion of cases in which the VLLM selects the caption with a higher level of hallucination $j$ over $k$. Specifically, we examine three key cases: when the most hallucinatory caption is chosen over both the lower-hallucination and anchor captions, and when the lower-hallucination caption is selected over the anchor caption. These cases are represented by $HM_{3 \to 1}$, $HM_{3 \to 2}$, and $HM_{2 \to 1}$, respectively, with results presented in Figure 7.

Our findings show that advanced VLLMs, such as VideoLLaMA2 (72B), can generally distinguish positive captions from severely halluci-

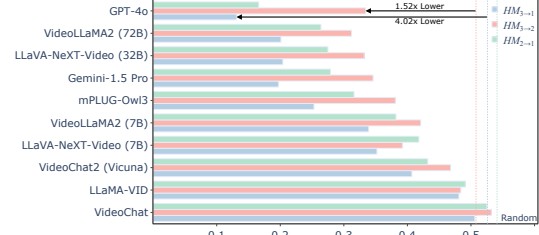

Figure 7: Hallucination misalignment (HM) scores on VIDHAL, with *Random* representing HM scores from the random baseline.

nated ones, as reflected by their low $HM_{3 \to 1}$ scores in Figure 7. However, two key observations emerge from our experiments: First, most VLLMs struggle to differentiate the lower hallucinatory caption from the anchor, as evidenced by the gap between $HM_{3 \to 1}$ and $HM_{2 \to 1}$. Second, all models exhibit high $HM_{3 \to 2}$ scores, indicating difficulty in distinguishing between two hallucinatory captions with varying degrees. These results suggests that gaps in nuanced video reasoning may contribute to hallucinatory behavior in VLLMs, a challenge not addressed by existing ⟨*positive*, *hallucinatory*⟩-based evaluation methods. [31, 53, 54, 15].

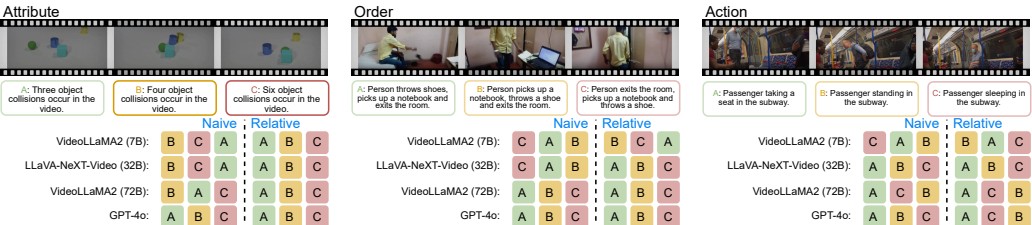

Figure 9: Qualitative examples of VLLM responses on the caption ordering tasks, for the *Attribute*, *Order* and *Action* aspects.

**Image Prior Reliance.** Previous research shows that VLLMs often rely on image priors for reasoning [21, 3], overlooking key spatiotemporal features. This is exemplified by a few frames having dominant influence on response generation. To examine how this bias affects hallucination generation in video-based VLLMs, we used a video summarization algorithm [48] to extract the most salient frame $v^i$ from $V^i$. We then generated VLLM responses on VIDHAL using $v^i$ instead of $V^i$ as the visual input. The effect of image priors is evaluated by identifying overlapping instances where responses from $V^i$ and $v^i$ remain consistent across both correct and incorrect orderings. As shown in Figure 8, results reveal that VLLMs heavily rely on image

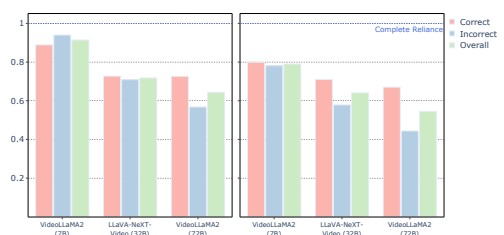

Figure 8: Overlapping ratios of model predictions under single-frame and full-video inputs for correct, incorrect and overall predictions. *Complete Reliance* indicates that the VLLM always produces the same response for both video and single frames.

priors. This is especially pronounced in smaller models such as VideoLLaMA2 (7B).

## 5.4 Qualitative Results

We conducted a qualitative analysis of responses generated by various VLLMs for the caption ordering task, with examples shown in Figure 9. We observe that: i) Relative caption ordering generally guides VLLMs to produce more accurate responses, as evidenced by improvements from naive to relative caption order predictions in most cases. ii) Advanced VLLMs exhibit more stable performance across both ordering tasks, with lower variation in predictions between both sub-tasks.

## 6 Conclusion

**Summary.** In this work, we introduce the VIDHAL benchmark to address gaps in the video-based hallucination evaluation of VLLMs. VIDHAL features video instances spanning five temporal aspects. Additionally, we propose a novel caption ordering evaluation task to probe the fine-grained video understanding capabilities of VLLMs. We conduct extensive experiments on VIDHAL through the evaluation of thirteen VLLMs, exposing their limitations in unexpected hallucination generation. Our empirical results shed light on several promising directions for future work: *e.g.*, incorporating a broader range of temporal features during pretraining and mitigating single-frame priors to enhance temporal reasoning. These advancements will help to address the hallucination problem in video-based VLLMs, enhancing their robustness for real-world video understanding applications.

**Limitations.** We acknowledge that the VIDHAL evaluation suite relies on synthetic captions generated by GPT-4o, which may contain biases inherently present in the model. We note that this design choice is consistent with prior research, as several established language-only and vision-language benchmarks similarly use GPT-4o for dataset construction [38, 24, 29, 23, 27] or response evaluation [15, 50, 32]. To reduce over-alignment to GPT-4o's preferences, we leverage additional strong LLMs, including Gemini-1.5 [46] and LLaMA2 (70B) [52] to assess and filter generated captions. While this improves the robustness of the annotations, we recognize that fully mitigating LLM-induced biases in caption generation remains an open challenge.

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
