# OpenReview forum: "VidHal: Benchmarking Hallucinations in Vision LLMs"
_NeurIPS.cc/2025/Datasets_and_Benchmarks_Track — Submitted to NeurIPS 2025 Datasets and Benchmarks Track_

### Official Review · Reviewer_NjK2 · 2025-06-21

**Rating:** 4
**Confidence:** 5

**Summary:**

- This work introduces an evaluation benchmark named VidHal, to study hallucination in MLLMs for video tasks.

- This dataset consists of 1000 videos sourced from existing benchmarks and paired with one correct and two hallucinatory captions and tasked MLLMs for 2 types of tasks: MCQ and caption ordering task. These captions are generated using GPT4o.

**Dataset Code Accessibility:**

Yes

**Dataset Code Comments:**

Code and benchmark are available.

**Ethical Considerations:**

No, there are no or only very minor ethics concerns

**Final Justification:**

After reading the authors' response to my post-rebuttal comments, I am increasing my score to **borderline accept**. I suggest that the authors incorporate the suggested changes to update the paper. My main remaining concern is that the authors have not provided results for top-performing models such as o1, o3, Gemini 2 Pro, and Gemini 2.5 Pro.

**Limitations Weaknesses:**

- The statement that “previously unexplored aspects such as directional movement” are studied in this work is incorrect, as several recent efforts have been made to specifically address this gap. Notable benchmarks such as TemporalBench, TVBench, TempCompass, and VideoHallucer have been introduced precisely for this purpose.

- Regarding Section 3.2 on Temporal Concept Selection: several of the subcategories listed under this section are incorrectly identified as temporal concepts. For instance, properties such as size, shape, color, and count relate to static object attributes rather than inherently temporal phenomena.

- A brief explanation is needed around line 115 to clarify how Sections 3.1, 3.2, and 3.3 are connected. It is unclear why understanding Sections 3.1 and 3.2 is necessary to follow the dataset construction described in Section 3.3. As it stands, the narrative flow is disjointed and should be improved.

- Line 166: Please elaborate on what kind of metadata is provided to GPT-4o as input for generating captions. It is unclear what source videos are used (beyond the mention of MVBench and AutoEval-Video), including their length, duration, and content. Based on the example shown in Figure 2, the videos appear overly simplistic, and the generated captions seem rather naive.

- There appears to be no scientific basis for classifying “The clock hands are moving counter-clockwise.” as a moderate hallucination and “The clock hands are stationary.” as a high hallucination. As such, the concept of assigning subjective degrees to hallucinations lacks rigor and appears neither scientifically grounded nor quantifiable.

- Line 172: The variables $j$ and $k$ are probably not defined.

- The process for generating hallucinatory captions appears uncontrolled. Simply prompting GPT-4o with “You are tasked with generating hallucinatory captions for a video with the description...” is insufficient. It is unclear how GPT-4o is expected to distinguish hallucinations solely based on the correct caption. The model might produce outputs that are entirely incorrect but do not qualify as reasonable hallucinations, or it could generate captions that differ from the original without being truly hallucinatory. For an example of more controlled hallucination generation, you may see this paper: Mitigating Object Hallucination in MLLMs via Data-augmented Phrase-level Alignment, ICLR 2025.

- Only 3-4 open-source models are studied and miss several recent open-source models of varying sizes, such as InternVL2.5, LongVU, MiniCPM, LongVILA, NVILA, and Qwen2.5VL; closed-source models Gemini 2 and o1. They should include diverse models from different architectures and sizes with diverse capacities.

- Additionally, using captions generated by GPT-4o and then evaluating GPT-4o on those same captions raises concerns about evaluation validity. This setup introduces a potential bias, which may partly explain why GPT-4o performs best on this benchmark. The authors can conduct additional experiments to substantiate that there is no such bias.

**Strengths Contributions:**

- I appreciated the authors' exploration of the caption preference ordering task, which offers an interesting aspect of evaluation in contrast to commonly used approaches like MCQ or BinaryQA setups.

- Hallucination in MLLMs is an important area of study, making this work both timely and relevant.

---

> ### Author Rebuttal · Authors · 2025-07-29
>
> We are grateful for the insightful feedback from the reviewer, and address the points raised below.
>
> - **The statement that “previously unexplored aspects such as directional movement” are studied in this work is incorrect, as several recent efforts have been made to specifically address this gap. Notable benchmarks such as TemporalBench, TVBench, TempCompass, and VideoHallucer have been introduced precisely for this purpose.**
>   We sincerely apologize for the confusion caused by this statement and appreciate the reviewer’s attention to this point. We would like to highlight that existing studies on video hallucinations have not explicitly evaluated aspects such as directional movement. Specifically, while benchmarks like TemporalBench, TVBench, and TempCompass have made significant contributions in evaluating temporal reasoning and consistency, they do not directly focus on hallucination detection or mitigation. VideoHallucer investigates both extrinsic and intrinsic hallucinations but does not explicitly consider fine-grained temporal elements such as directional or trajectory-based hallucinations during its benchmark construction.
>
>     Our work aims to bridge this gap by incorporating these directional and temporal aspects as a dedicated evaluation dimension, complementing and extending the scope of prior efforts.
> - **Regarding Section 3.2 on Temporal Concept Selection: several of the subcategories listed under this section are incorrectly identified as temporal concepts. For instance, properties such as size, shape, color, and count relate to static object attributes rather than inherently temporal phenomena.**
>   We sincerely thank the reviewer for this insightful comment. The inclusion of these spatial properties under Temporal Concept Selection is deliberate, as we evaluate these attributes in the context of their **dynamic changes over time**.
>
>     For example, attributes like color or count can vary temporally, such as a traffic light changing from green to red or the number of vehicles changing as they enter or leave a scene. We illustrate more examples in Figure 26 of the supplementary material. These cases require temporal reasoning to accurately track attributes across different time points and identify their transitions.
>
> - **A brief explanation is needed around line 115 to clarify how Sections 3.1, 3.2, and 3.3 are connected. It is unclear why understanding Sections 3.1 and 3.2 is necessary to follow the dataset construction described in Section 3.3.**
>   We greatly appreciate the reviewer’s observation and apologize for the lack of clarity in the narrative flow. Sections 3.1 and 3.2 are intended to establish the conceptual foundation for our dataset construction. Specifically, Section 3.1 defines how hallucinations may manifest in model-generated responses under video inputs, while Section 3.2 categorizes the dimensions of temporal hallucinations that are most critical for assessing VideoLLMs. These sections collectively and comprehensively outline the types of hallucinations and failure modes we aim to capture in existing models. These definitions guide the systematic selection and annotation of video instances in Section 3.3, ensuring that the dataset comprehensively addresses the key types of hallucinations. We will revise this section to make these connections more explicit and to improve the narrative flow in our paper.
>
> - **Line 166: Please elaborate on what kind of metadata is provided to GPT-4o as input for generating captions. It is unclear what source videos are used (beyond the mention of MVBench and AutoEval-Video), including their length, duration, and content. Based on the example shown in Figure 2, the videos appear overly simplistic, and the generated captions seem rather naive.**
>   We sincerely thank the reviewer for highlighting this point and apologize for the lack of clarity in the paper. For TempCompass, MVBench, and PerceptionTest, we provide GPT-4o with the QA pairs associated with each video instance as metadata. For AutoEval-Video, the long-form human-written captions accompanying the videos are supplied as input. These annotations serve as semantic anchors for the evaluated temporal concepts, guiding GPT-4o to focus on the core aspects relevant to the hallucination benchmark.
>
>     While the examples in Figure 2 may appear simplified, they were intentionally chosen for clarity and to illustrate the behavior of the evaluated models. The full benchmark encompasses videos with varying complexity, including synthetic, acted, and open-domain natural scenes, to thoroughly assess temporal reasoning and fine-grained understanding. The videos in _VidHal_ have an average duration of 15.8 seconds and an average caption length of 11.2 words. A detailed distribution of video durations and caption lengths is provided in the supplementary material. We kindly refer the reviewer to the respective dataset papers for detailed descriptions of video collection, duration distributions, and content diversity.
>
> - **There appears to be no scientific basis for classifying “The clock hands are moving counter-clockwise.” as a moderate hallucination and “The clock hands are stationary.” as a high hallucination. As such, the concept of assigning subjective degrees to hallucinations lacks rigor and appears neither scientifically grounded nor quantifiable.**
>   We appreciate the reviewer’s observation and the opportunity to clarify our rationale for the hallucination levels. In our framework, hallucination levels are defined relative to an anchor caption, where the level increases with the degree of contradiction or divergence from the anchor. For example, if the anchor states, “The clock hands are moving clockwise,” the caption “The clock hands are moving counter-clockwise” represents a moderate hallucination, as the direction is incorrect but the action (movement of clock hands) is preserved. In contrast, “The clock hands are stationary” constitutes a high hallucination because it contradicts both the direction and dynamics described in the anchor, misrepresenting the core scene.
>
> - **Line 172: The variables $j$ and $k$ are probably not defined.**
>   We appreciate the reviewer’s keen observation. We acknowledge that the variables $j$ and $k$ were not explicitly defined in the current version of the manuscript. We will revise the paper to include clearer definitions of these variables.
>
> - **The process for generating hallucinatory captions appears uncontrolled. Simply prompting GPT-4o with “You are tasked with generating hallucinatory captions for a video with the description...” is insufficient. It is unclear how GPT-4o is expected to distinguish hallucinations solely based on the correct caption.**
>   We thank the reviewer for this insightful comment and for highlighting relevant work. We will draw insights from this work and cite it in the revised manuscript. To ensure controlled hallucination generation, we define each hallucination level with clear, precise criteria and provide GPT-4o with structured instructions, supplemented by five in-context examples for each aspect and sub-aspect (see Figures 17 to 23). These measures guide the model toward producing intentional and meaningful deviations rather than arbitrary errors. Furthermore, each step of the caption generation pipeline is manually reviewed and refined to ensure high quality of intermediate outputs within the annotation pipeline. We appreciate and acknowledge suggestions for further enhancements by the reviewer, and will strongly consider exploring such approaches in future iterations of _VidHal_.
>
> - **Only 3-4 open-source models are studied and miss several recent open-source models of varying sizes, such as InternVL2.5, LongVU, MiniCPM, LongVILA, NVILA, and Qwen2.5VL; closed-source models Gemini 2 and o1.**
>   We sincerely thank the reviewer for highlighting this point and for their valuable suggestion. In addition to the six open-sourced model families evaluated in our paper, we have expanded our experiments to include an extensive range of the mentioned models, covering a broader range of architectures and capacities. The additional results are presented in Table 5, and will be included in our revised paper.
>   | Model                  | Accuracy | NDCG (Naive) | NDCG (Relative) |
>   |------------------------|----------|--------------|-----------------|
>   | InternVL2.5 (8B)       | 0.773    | 0.475        | 0.827           |
>   | InternVL2.5 (26B)      | 0.742    | 0.498        | 0.775           |
>   | LongVU (8B)            | 0.795    | 0.453        | 0.846           |
>   | MiniCPM (7B)           | 0.377    | 0.530        | 0.523           |
>   | Qwen2.5-VL (7B)        | 0.760    | 0.825        | 0.826           |
>   | Qwen2.5-VL (26B)       | 0.732    | 0.811        | 0.800           |
>
>   **Table 1: Benchmark performance of additional VLLMs on our _VidHal_ benchmark.**
> - **Using GPT-4o-generated captions for evaluating GPT-4o raises validity concerns, as this setup may introduce bias and partly explain its superior performance on the benchmark.**
>   Thank you for raising this important concern regarding potential bias from using GPT-4o-generated captions. To mitigate this, we deliberately employ the text-only variant of GPT-4o to avoid vision-based priors that could artificially inflate hallucination evaluation performance. In addition, we incorporate multiple independent LLMs (Gemini-1.5 and LLaMA2-70B) to filter and validate the captions, reducing the risk of overfitting or alignment bias toward GPT-4o. This multi-model pipeline is designed to ensure that our benchmark robustly measures hallucination in VideoLLMs rather than reflecting the preferences of a single model (GPT-4o).
>
>     We fully agree on the importance of further validating potential biases, and plan to explore concrete approaches to validate the presence and extent of such biases in future versions of _VidHal_.

---

> > ### Comment · Reviewer_NjK2 · 2025-08-04
> > **post-rebuttal comments**
> >
> > I thank the authors for responding to my comments.
> >
> > - "... existing studies ... have not explicitly evaluated aspects such as directional movement ..." → The way video-QA benchmarks like TemporalBench, TVBench, TempCompass, and VideoHallucer are constructed in the literature, the distinction between hallucination and non-hallucination is often subtle and overlapping. Some papers frame this as fine-grained temporal understanding vs. temporal hallucination, depending on their broader scope. Therefore, it’s unclear how your work meaningfully differentiates itself. Furthermore, since your benchmark is based on TempCompass and MVBench/TVBench (noting that TVBench itself is based on MVBench), it further supports the point that the novelty or distinctiveness of your benchmark is limited.
> >
> > - "Specifically, ... TemporalBench, TVBench, and TempCompass ..., they do not directly focus on hallucination detection or mitigation." → But your benchmark does not focus on hallucination mitigation either, so why bring this up?
> >
> > - It is still unclear from the textual description in the paper how changes in spatial aspects are broadly included in the dataset. Also, I couldn’t find Figure 26 in the supplementary material as referenced by the authors, the figures go up to Figure 23.
> >
> > - "we provide GPT-4o with the QA pairs associated with each video instance as metadata" → This setup for generating captions seems flawed. A better approach would be to provide richer ground-truth information, such as a detailed object/action lists and their boudning boxes. Otherwise, the generated captions may lack specificity or, worse, include hallucinated content that coincidentally matches some ground-truth elements. Since this is meant to be an evaluation benchmark, the setup needs to be more foolproof.
> >
> > - Classifying hallucination severity (i.e., more vs. less hallucination) is highly subjective, unless you go with quantitive measures, which is not the case here. Ranking models based on such subjective assessments may not be appropriate.
> >
> > - Many key models are still missing from your benchmarking. The authors did not include several top-performing models -- both closed-source (e.g., Gemini 2, Gemini 2.5, o1, o3) and open-source (e.g., Qwen 2.5 VL 72B, InternVL2.5 78B) -- even after being explicitly asked.
> >
> > - Using GPT-4o in a text-only setup for caption generation seems problematic. It may introduce strong linguistic biases, especially since it lacks visual grounding. For a more robust and trustworthy evaluation, there should be experiments demonstrating that judgments across models correlate well with each other and with human evaluations.

---

> > > ### Author Response · Authors · 2025-08-05
> > > **Response to Post-Rebuttal Comments**
> > >
> > > We thank the reviewer for the active discussion, and would like to further address the concerns raised below.
> > >
> > > - **Limited novelty and distinctiveness of _VidHal_.**
> > >   We thank the reviewer for the keen observation and thorough study of existing video understanding benchmarks. Firstly, we would like to clarify that `fine-grained temporal understanding and temporal hallucinations are not equivalent`. The former assesses a model’s ability to capture precise temporal cues, while the latter focuses on identifying instances where the model produces temporally inaccurate outputs that appear plausible. _VidHal_ is explicitly designed to surface such hallucinations, which may go undetected in standard Video QA settings. Second, _VidHal_ distinguishes from these benchmarks in two ways:
> > >   1.  While we leverage visual instances from existing benchmarks, we do not reuse their annotations. Instead, we create a novel set of captions exhibiting varying degrees of temporal hallucinations, which is absent in prior work.
> > >   2. _VidHal_ introduces a novel caption ordering task that evaluates a model’s ability to rank captions by hallucination degree. This presents a more complex challenge than the binary or multiple-choice formats of existing benchmarks, revealing limitations in current VideoLLMs not captured by prior evaluation settings.
> > >
> > > - **Mention of hallucination mitigation is unclear, as it is not directly addressed in _VidHal_**
> > >   We thank the reviewer for raising this concern. Our intention was to clarify that while benchmarks such as TemporalBench, TVBench, and TempCompass focus on evaluating temporal understanding in VideoLLMs, they do not explicitly address the challenge of hallucinations arising from video inputs. In contrast, `_VidHal_ is specifically designed to evaluate and analyze hallucinations` by revealing systematic failure modes across diverse video scenarios. We believe this targeted focus distinguishes _VidHal_ and serves as a crucial first step toward enabling the development of effective mitigation strategies for video hallucinations.
> > > - **The inclusion of spatial aspect changes in _VidHal_ remains unclear, and Figure 26 referenced in the supplementary material could not be found.**
> > >   We sincerely apologize for the oversight in our initial rebuttal. We would like to correct this by directing the reviewer to Figure 17 of the supplementary material. Specifically, the first two rows of Figure 17 illustrate examples from _VidHal_ involving object and attribute aspects, highlighting how these spatial attributes evolve over time.
> > > - **Absence of grounding information such as bounding boxes causes generated captions to lack specificity.**
> > >   Thank you for your thoughtful response. We agree that incorporating rich grounding information, such as object/action lists and bounding boxes, can reduce the risk of misaligned annotations. However, our choice to use the metadata is driven by _VidHal_’s objective to evaluate hallucinations across a broader spectrum of temporal aspects. While spatial attributes like objects can be supported by additional annotations, extending such labels to capture temporal concepts, such as event order and directional cues, is non-trivial. To minimize the risk of misleading metadata, we have applied a rigorous filtering and verification process prior to our annotation pipeline, retaining only videos where the metadata reliably anchor the semantics of the underlying temporal dynamics.
> > > - **Many key models, such as Qwen 2.5 VL and InternVL2.5, are still missing from the benchmarking.**
> > >   We appreciate the reviewer’s emphasis on comprehensive benchmarking. In response to earlier feedback, `we have incorporated evaluations of Qwen2.5-VL and InternVL2.5 across multiple model sizes, among other open-sourced VideoLLMs, as shown in Table 1 of the rebuttal`. For proprietary models such as Gemini 2, Gemini 2.5, o1, and o3, benchmarking was unfortunately not feasible within the rebuttal period due to inference costs and time constraints. Nonetheless, we recognize the importance of including these models and are committed to evaluating them in future iterations of _VidHal_.
> > > - **The use of text-only GPT-4o in caption generation may induce linguistic biases in the benchmark.**
> > >   We sincerely appreciate the reviewer’s insightful comment regarding the potential linguistic biases of using text-only GPT-4o in caption generation. To minimize this risk, we employ a two-step quality control procedure. First, we reduce dependence on GPT-4o by leveraging additional powerful LLMs, Gemini-1.5 and LLaMA2 (70B), to evaluate and filter GPT-4o’s generated captions, thus mitigating individual model biases. Next, we conduct a final round of human verification and editing to address any residual misalignments not caught by automated filtering. These steps ensure the production of high-quality, reliable captions that accurately reflect the visual content and its associated temporal aspect.

---

### Official Review · Reviewer_dETu · 2025-07-01

**Rating:** 4
**Confidence:** 4

**Summary:**

VIDHAL is an interesting effort to create a benchmark specifically for evaluating video-based hallucinations in Vision Large Language Models (VLLMs). The benchmark addresses the limitations of existing research by focusing on introducing a new task which relies on detecting or evaluating the order of generated content. VIDHAL is constructed using video instances that cover a wide range of temporal aspects and includes captions representing varying levels of hallucination. It employs GPT4-o to generate the captions. A randomly selected instances of data is validated by human to ensure the automatic generation of dataset is of high quality.

**Dataset Code Accessibility:**

Yes

**Dataset Code Comments:**

The code and data both seems available and could not spot an issue accessing.

**Ethical Considerations:**

No, there are no or only very minor ethics concerns

**Final Justification:**

Considering the discussions and the response from authors, I find the paper an interesting one. I believe it has merit to be considered and I am more confident in leaning toward accepting it.

**Limitations Weaknesses:**

* The reliance on synthetic captions generated by GPT-4o may introduce biases inherent in the model, which is acknowledged as a limitation.
* using just a handful of validated examples by human could cast doubts a bit on the quality and existence of outlier instances.

**Strengths Contributions:**

To summarize, there are several aspects that are interesting with regard to current paper:

* The paper addresses a significant gap in the evaluation of video-based hallucinations in VLLMs, which is an underexplored area, IMO.
* VIDHAL benchmark tries to be comprehensive. To this end, it tries to cover a broad range of temporal aspects and providing multiple levels of hallucination for evaluation.
* The proposed task of caption ordering allows assessment of hallucinations compared to binary question-based benchmarks.
* Extensive experiments and evaluations are conducted, providing valuable insights into the limitations of current VLLMs.

---

> ### Author Rebuttal · Authors · 2025-07-29
>
> We thank the reviewer for the favorable feedback and would like to address the concerns outlined below.
>
> - **The reliance on synthetic captions generated by GPT-4o may introduce biases inherent in the model, which is acknowledged as a limitation.**
>   We thank the reviewer for highlighting this important limitation that synthetic captions from GPT-4o may introduce biases. We aim to reduce the inherent preferences of GPT-4o, by using the text-only variant of GPT-4o to avoid vision-based priors and additional LLMs (Gemini-1.5 and LLaMA2 (70B)) to further filter and validate the generated captions. This design helps ensure that our evaluation robustly captures hallucination in VideoLLMs while limiting the effect of biases on hallucination evaluation.
>
> - **Using just a handful of validated examples by human could cast doubts a bit on the quality and existence of outlier instances.**
>   We appreciate the reviewer’s concern and agree that a larger validation set could further enhance the robustness of our annotation pipeline. Our choice of 100 samples (10\% of the benchmark) was a deliberate trade-off between ensuring sufficient coverage for quality control and `managing annotation costs`. Importantly, these samples were carefully selected to span all temporal aspects and sub-aspects that _VidHal_ is designed to evaluate, ensuring that the validation subset is representative of our benchmark dataset.

---

### Official Review · Reviewer_YFFc · 2025-07-03

**Ethics Flags:** Data privacy, copyright, and consent
**Rating:** 4
**Confidence:** 4

**Summary:**

The submission presents VIDHAL, a 1000-video benchmark for diagnosing temporal hallucinations in vision-language models (VLLMs). Five temporal aspects (Action, Attribute, Direction, Object interaction and Event Order) are covered. Each clip is paired with an anchor caption and two GPT-4o-generated hallucinated captions of increasing severity. Two evaluation tasks are introduced: (i) a multiple-choice QA that asks models to choose between an anchor and one hallucination, and (ii) caption-ordering, where models must rank the three captions by faithfulness. Thirteen public and proprietary VLLMs are benchmarked; all struggle, particularly on Direction and Order, revealing a pronounced single-frame bias.

**Additional Feedback:**

The study tackles an important, under-explored problem and introduces a thoughtful evaluation protocol, yet its modest scale, heavy reliance on synthetic captions, and limited human vetting dilute its impact; expanding the dataset, adding human-authored labels, and increasing validation coverage would markedly strengthen both rigor and practical utility.

**Dataset Code Accessibility:**

Yes

**Dataset Code Comments:**

Overall well-structured. Add a LICENSE (e.g., Apache-2.0).

**Ethical Comments:**

The benchmark republishes clips from four external datasets under heterogeneous licences. Explicit confirmation of redistribution rights and privacy safeguards is needed.

**Ethical Considerations:**

Yes, there are ethics concerns that require attention by the authors

**Final Justification:**

The reviewer feels that there are still areas to improve, but does not consider it a critical flaw that would preclude acceptance, thus learning toward acceptance.

**Limitations Weaknesses:**

[W1] The benchmark includes only 1000 clips no longer than 16 s and sourced from four datasets, omitting egocentric, surveillance, and cinematic footage, which limits external validity. Expanding the corpus with longer, more diverse real-world video sources (e.g., Ego4D, CityFlow, movie trailers) would better stress temporal reasoning across domains.

[W2] Anchors and hallucinations are generated and filtered entirely by LLMs, which risks shared-bias leakage between the dataset and the models being evaluated.

[W3] Caption ordering is evaluated only with NDCG, which penalizes a moderate-to-severe swap the same as an anchor-to-severe swap, and no alternative rank metrics or sensitivity study is provided.

[W4] The single-frame bias analysis relies on one saliency algorithm and a fixed clip length, with no exploration of alternative summarization methods or longer-clip ablations. Testing multiple summarization strategies (e.g., uniform sampling, motion energy) across varied clip lengths would strengthen conclusions about temporal reasoning weaknesses.

**Strengths Contributions:**

[S1] While prior work is largely image-centric, VIDHAL targets temporal failure modes, offering timely insight into video-specific hallucinations in VLLMs.

[S2] The dual-task design, consisting of binary QA plus caption ordering, yields graded signals that reveal subtle temporal errors often missed by single-choice evaluations.

[S3] Compared with VidHalluc and VideoHallucer, VIDHAL broadens coverage by adding a Direction aspect and splitting Attributes into size, color, count, and state sub-categories.

[S4] A comprehensive evaluation of 13 open- and closed-source VLLMs, along with a single-frame ablation, exposes a pervasive frame bias and provides concrete targets for future model improvement.

---

> ### Author Rebuttal · Authors · 2025-07-29
>
> We express our sincere gratitude to the reviewer for the insightful feedback. We provide our responses to the comments presented by the reviewer in more detail as follows.
>
> - **The benchmark includes only 1000 clips no longer than 16 s and sourced from four datasets, omitting egocentric, surveillance, and cinematic footage, which limits external validity.**
>   Thank you for this insightful comment and suggestion. Our current selection of the four datasets (TempCompass, AutoEval-Video, MVBench, and PerceptionTest) was guided by their explicit coverage of the temporal concepts which our study focused on, such as event order and direction. These datasets enable us to curate concept-specific instances, up to 1 minute long, covering diverse temporal phenomena for a focused and comprehensive hallucination evaluation. We agree that incorporating egocentric, surveillance, and cinematic footage would further enhance both the visual and temporal diversity of _VidHal_, and will strongly consider this valuable suggestion to enrich and expand the benchmark in future iterations.
>
> - **Anchors and hallucinations are generated and filtered entirely by LLMs, which risks shared-bias leakage between the dataset and the models being evaluated.**
>   We thank the reviewer for raising this important concern. While human annotation is the gold standard, its cost is prohibitive, especially given the multiple levels of hallucination present in _VidHal_. To circumvent this challenge while ensuring the robustness of our benchmark, we conducted human validation on a subset of _VidHal_ to demonstrate the robustness of our automatic annotation pipeline and its strong alignment with human judgment.
>
> - **Caption ordering is evaluated only with NDCG, which penalizes a moderate-to-severe swap the same as an anchor-to-severe swap, and no alternative rank metrics or sensitivity study is provided.**
>   We thank the reviewer for this insightful comment and address it in two parts:
>   - **On NDCG penalizing swaps equally:**  We apologize for the confusion, would like to clarify that NDCG `does not` penalize a moderate-to-anchor swap the same as an anchor-to-severe swap. In our evaluation, NDCG assigns a heavier penalty when a severely hallucinated caption $y^{i,k}\_-$ (where $k > j$) is ranked above the anchor caption $y^i\_+$ compared to when a moderately hallucinated caption $y^{i,j}\_-$ is ranked above $y^{i}\_+$. For example, the orderings $\\{y^{i,1}\_-, y^{i}\_+, y^{i,2}\_-\\}$ (moderate-to-anchor swap) and $\\{y^{i,2}\_-, y^{i,1}\_-, y^{i}\_+\\}$ (severe-to-anchor swap) receive NDCG scores of $0.631$ and $0.00$, respectively, under our metric computation for caption ordering tasks in _VidHal_.
>   - **On alternative metrics and sensitivity studies**: Other established retrieval metrics such as $\text{MAP}@K$ and $\text{MRR}@K$ share similar concepts with NDCG, yet require an extensive definition of penalty. Following existing research in the information retrieval domain [_ColPali: Efficient Document Retrieval with Vision Language Models (ICLR 2025), DiSCo: LLM Knowledge Distillation for Efficient Sparse Retrieval in Conversational Search (SIGIR2025)_], we adopted NDCG as a representative that incorporates **position sensitivity** when evaluating retrieved elements, aligning with our goal of fine-grained hallucination assessment via caption ordering.
>
> - **The single-frame bias analysis relies on one saliency algorithm and a fixed clip length, with no exploration of alternative summarization methods or longer-clip ablations.**
>   We sincerely thank the reviewer for their constructive comment and suggestions. In our original experiments, we adopted the clip lengths used by each model (8 frames for VideoLLaMA (72B) and 32 frames for LLaVA-NeXT-Video (32B)) to preserve their intended sampling rates. We conduct additional single-frame bias experiments using two alternative sampling strategies, uniform and motion-based, along with ablations across shorter clip lengths (1, 2, and 4 frames). The results of these extended analyses are presented below.
>   | Model                   | 1 Frame (C/I/O)   | 2 Frames (C/I/O)   | 4 Frames (C/I/O)   |
>   |------------------------|------------------|--------------------|--------------------|
>   | VideoLLaMA2 (7B)       | 0.674 / 0.708 / 0.700 | 0.781 / 0.798 / 0.794 | 0.846 / 0.829 / 0.833 |
>   | LLaVA-NeXT-Video (32B) | 0.680 / 0.57 / 0.62   | 0.735 / 0.649 / 0.688 | 0.831 / 0.706 / 0.763 |
>
>   **Table 1: Overlapping ratios of model predictions under summarized and full-video inputs for (C)orrect, (I)ncorrect and (O)verall predictions across multiple sampling rates using uniform sampling**
>
>   The results demonstrate that the overlap ratios for single-frame inputs under both uniform and motion-based sampling are consistent with the saliency-based summarization used in our study (Figure 8). These findings confirm that our single-frame bias findings are robust to the summarization method, with many VLLMs relying on single-frame information for more than half of the queries in _VidHal_.

---

> > ### Comment · Reviewer_YFFc · 2025-08-04
> >
> > Thank you for the response.
> >
> > 1. While the reviewer acknowledges that conducting experiments on additional datasets (e.g., egocentric, surveillance, and cinematic footage) may not be feasible within the limited timeframe of the review process, these video domains represent long-standing and significant challenges in the field. Explicitly acknowledging this as a limitation and suggesting it as a direction for future work could help inspire further research and broaden the impact of the proposed approach.
> >
> > 2. The reviewer concurs that the alignment between human and LLM-based evaluations can serve as a proxy for human judgment. However, it is important to note that both the anchor and hallucinatory captions are generated by GPT-4o, and thus inherently reflect the model’s learned biases. These biases may not be fully captured through human evaluation alone. Interpreting the proposed method with this consideration in mind would strengthen its validity. While this aspect is noteworthy, the reviewer does not consider it a critical flaw that would preclude acceptance.

---

### Official Review · Reviewer_YZRP · 2025-07-03

**Rating:** 5
**Confidence:** 3

**Summary:**

This paper created a benchmark to evaluate video hallucination in large VLMs, considering multiple aspects of hallucinations such as order, attributes, and actions to capture temporal concepts. The authors constructed an evaluation dataset using four public video datasets containing a total of 1000 videos and introducing hallucinatory captions at different levels. They evaluated video hallucination in 13 VLMs using their two proposed metrics: MCQA and caption ranking.

**Dataset Code Accessibility:**

Yes

**Dataset Code Comments:**

The code and dataset is accessible.

**Ethical Comments:**

Authors have relied on existing publicly available video dataset to construct their dataset.

**Ethical Considerations:**

No, there are no or only very minor ethics concerns

**Final Justification:**

The reviewers have addressed my concerns and I still think this paper is a good contribution and would like to keep my score.

**Limitations Weaknesses:**

$\bullet$ The authors generated synthetic hallucinated captions using only GPT-4o, which could bias the evaluation toward GPT-4o-like responses. GPT-4o's score is the highest in Table 2, potentially because GPT-4o might have learned to identify certain shortcut patterns in its own caption wording. A more robust approach would have involved using multiple VLMs for generating synthetic hallucinations or directly utilizing real-world hallucinatinatory texts generated by VLMs. The authors have also acknowledged this limitation of a single-model approach to creating synthetic hallucinations in the conclusion section.


$\bullet$ Though the code is included, the authors could discuss more about inference settings of used models, especially open-sourced ones, in supplementary section. Also, please report time required for inference.

**Strengths Contributions:**

$\bullet$ This benchmark evaluates hallucinations at different levels and covers several temporal aspects such as order and direction, which makes the evaluation strong.

$\bullet$ The proposed NDCG metric is well thought out and credits for partially correct ordering.

$\bullet$ The additional analysis at the aspect level provides good insight into where the models fail.

$\bullet$ The overall paper is well-structured with clear sections and provides important data statistics.

$\bullet$ Includes human assessment to validate the alignment of synthetically introduced hallucination in the dataset to human perception.

---

> ### Author Rebuttal · Authors · 2025-07-29
>
> We thank the reviewer for the constructive feedback and positive review of our paper. We address the individual comments raised by the reviewer below.
>
> - **Using only GPT-4o to generate synthetic hallucinated captions may bias the evaluation toward GPT-4o-like responses, possibly due to recognizing shortcut patterns in its own captions.**
>
>   We thank the reviewer for raising this important point. We acknowledge the potential bias toward GPT-4o due to using it for generating synthetic hallucinated captions. To mitigate such bias, we employed the text-only variant of GPT-4o for caption generation, ensuring no vision-based information influenced the synthetic data, which were then filtered by additional LLMs (Gemini-1.5 and LLaMA2 (70B)). Moreover, we carefully and manually selected and annotated video instances that require a holistic understanding of video dynamics, reducing the influence of language-only biases on hallucination evaluation.
> - **Though the code is included, the authors could discuss more about inference settings of the used models, especially open-sourced ones, in supplementary section. Also, please report time required for inference.**
>
>   We thank the reviewer for this helpful suggestion. We have added further details of the inference and generation settings used across all evaluated models in Table 1.
>
>
>   | Hyperparameter             | Value      |
>   |---------------------------|------------|
>   | **_Data Processing_**     |            |
>   | Video Sampling Rate (FPS) | 30         |
>   | **_Generation_**          |            |
>   | `do_sample`               | `False`    |
>   | `temperature`             | `0.0`      |
>   | `repetition_penalty`      | `1.0`      |
>   | `max_new_tokens`          | `128`      |
>   | **_Computation_**         |            |
>   | Precision                 | FP16       |
>   **Table 1: Additional model-agnostic inference hyperparameters**
>
>   For the LLaVA-NeXT-Video models, we additionally provide model-specific architectural configurations for each variant in Table 2.
>
>   | Hyperparameter             | LLaVA-NeXT-Video (7B) | LLaVA-NeXT-Video (32B) |
>   |---------------------------|------------------------|-------------------------|
>   | `mm_spatial_pool_mode`    | `average`              | `average`               |
>   | `mm_newline_position`     | `no_token`             | `grid`                  |
>   | `mm_pooling_position`     | `after`                | `after`                 |
>   **Table 2: Model-specific hyperparameters for LLaVA-NeXT-Video**
>
>    Table 3 reports the average inference time per sample on our _VidHal_ benchmark for all three tasks, with relative caption ordering incurring the longest duration due to the use of multiple binary QA prompts.
>
>
>
>   | Model                      | MCQA | Naive Ordering | Relative Ordering |
>   |---------------------------|------|----------------|-------------------|
>   | LLaMA-VID                 | 5.9  | 4.1            | 8.7               |
>   | VideoChat2 (Vicuna)       | 6.2  | 4.3            | 8.8               |
>   | VideoChat2 (Mistral)      | 11.2 | 9.7            | 16.3              |
>   | VideoChat2 (Phi)          | 7.9  | 8.0            | 10.9              |
>   | mPLUG-Owl3                | 6.6  | 5.1            | 8.4               |
>   | LLaVA-NeXT-Video (7B)     | 5.1  | 4.0            | 6.7               |
>   | LLaVA-NeXT-Video (32B)    | 4.9  | 3.8            | 6.4               |
>   | VideoLLaMA2 (7B)          | 4.3  | 4.1            | 5.7               |
>   | VideoLLaMA2 (72B)         | 4.7  | 4.4            | 5.9               |
>   **Table 3: Inference time (s/sample) for different models on _VidHal_**
>
>
>   These details will also be included in the supplementary section of the revised paper.

---

### Note · Authors · 2025-08-13

We thank all reviewers for their constructive feedback, and active discussion. We briefly restate the strengths of _VidHal_ noted by the reviewers and summarize the key concerns raised along with our responses.

**Strengths**:
- _VidHal_ presents a focused benchmark for temporal hallucinations in VideoLLMs with a comprehensive coverage of temporal aspects in its evaluation.
- _VidHal_ introduces a novel caption ordering task to complement binary QA, allowing a more fine-grained assessment of temporal hallucinations.
- Extensive experiments have been conducted on _VidHal_, with detailed studies that reveal potential failure modes of existing VideoLLMs.

**Concerns and Responses:**
- **Bias from GPT-4o caption generation (all reviewers)**, We mitigated potential biases via three key steps:
  - Carefully crafted instructions, comprehensive guidelines, and aspect-specific context examples.
  - Additional validation and filtering using Gemini-1.5 and LLaMA2 (70B).
  - Manual checks, filtering and edits alongside human validation on a subset of _VidHal_.

- **Limited ranking metrics studies (YFFc)**: We clarified that NDCG’s position-sensitivity aligns well with our goal of penalizing more severe misorderings more heavily, making it more suitable over other metrics such as MAP and MRR.

- **Single-frame bias analysis (YFFc)**: We provide additional ablations using alternative summarization strategies and varying clip lengths checks, with results supporting the single-frame reliance reported in the paper, confirming the robustness of our analysis.

- **Inclusion of spatial aspects (NjK2)**: We clarified that spatial concepts (objects, attributes) are evaluated in their temporal context, such as how they change over time, rather than as static properties.

- **Metadata and statistics (NjK2)**: We elaborate on the metadata supplied to GPT-4o and reported dataset statistics, subsequently referring the reviewer to the supplementary material for full details.

- **Model coverage (NjK2)**: We extended our evaluation by adding six additional open-source models across multiple sizes and four families.

- **Overlap with prior benchmarks (NjK2)**: We noted that while existing benchmarks assess similar temporal aspects of video understanding, they do not systematically examine hallucination cases, a gap that _VidHal_ addresses.

We again thank the reviewers for their insightful suggestions, and will strongly consider incorporating them in future iterations of _VidHal_.

---

### Decision · Program_Chairs · 2025-09-18

**Decision:**

Reject

**Comment:**

some concerns were raised by the reviewers which were answered in a good way by the authors in the rebuttal. There is an agreement among the reviewers that the paper should be accepted.

===== FINAL UPDATE FROM DB Track PCs ====

The final decision for this paper has been taken by the program chairs after consultation with the SACs. All Senior Area Chairs have ranked papers according to the feedback from the AC during the review process. We decided to leave the original meta-review to reflect the opinion of the AC in light of the initial discussions with reviewers and SAC.